# Association between Psychological Distress and Possible, Probable, and Definite Sleep Bruxism—A Comparison of Approved Diagnostic Procedures

**DOI:** 10.3390/jcm13020638

**Published:** 2024-01-22

**Authors:** Nicole Pascale Walentek, Ralf Schäfer, Nora Bergmann, Michael Franken, Michelle Alicia Ommerborn

**Affiliations:** 1Department of Operative Dentistry, Periodontology, and Endodontology, Faculty of Medicine, Heinrich-Heine-University, Moorenstr. 5, 40225 Düsseldorf, Germanymichelle.ommerborn@ommerborn.net (M.A.O.); 2Clinical Institute of Psychosomatic Medicine and Psychotherapy, Faculty of Medicine, Heinrich-Heine-University, Moorenstr. 5, 40225 Düsseldorf, Germany

**Keywords:** sleep bruxism, psychological distress, polysomnography, clinical assessment, self-report

## Abstract

(1) Background: The relationship between sleep bruxism (SB) and psychological distress has been investigated in numerous studies and is heterogeneous. Different diagnostic procedures have been applied to determine SB. The aim of this study was to directly compare the association between psychological distress and SB diagnosed by different accepted methods. (2) Methods: Data of *N* = 45 subjects were analyzed, including group comparisons and correlation analyses. Following diagnostic methods for the determination of SB were used in one sample: self-report, clinical assessment, polysomnography with audio–video recording and a novel diagnostic sheet with analyzing software. Psychological distress was measured using the global severity index (GSI) of the Symptom Checklist-90-Standard (SCL-90-S). (3) Results: The GSI did not differ significantly between subjects with and without SB, regardless of the underlying diagnostic classification (*p* > 0.05). In-depth correlation analyses of self-report and clinical data revealed a weak-to-medium correlation with the GSI (r = 0.12–0.44). Due to non-normally distributed data, a test of statistical significance was not possible. Variables of instrumental methods such as the SB index (amount of SB activity per hour) of polysomnography (PSG) showed almost no correlation with psychological distress (r = −0.06–0.05). (4) Conclusions: Despite these limitations, the results provide an indication that the choice of diagnostic procedure may elucidate the variance in the correlation between SB and psychological distress.

## 1. Introduction

Sleep bruxism is a masticatory muscle activity during sleep that is characterized as rhythmic (phasic) or non-rhythmic (tonic) and is not a movement disorder or a sleep disorder in otherwise healthy individuals [1]. The intensity of SB activity can be viewed on a continuum, with the presence of an extremely frequent and untreated form of SB leading to disorders. In adults, the prevalence of SB ranges from 5.5 to 15.0% depending on the underlying diagnostic method [2,3,4].

The etiology of SB is multifactorial [1]. Certain neurotransmitters, such as serotonin, or specific genetic conditions have been discussed at the biological level [5,6]. Diseases like Parkinson’s disease and obstructive sleep apnea syndrome are associated with an increased rate of SB [7,8]. Exogenous factors such as tobacco use are positively related to SB [9]. Numerous studies show heterogeneous results concerning an association between SB and psychosocial factors such as stress and maladaptive management of stress, negative emotions, or mental illness [10,11,12,13,14]. How SB is associated with the development of painful temporomandibular disorders (TMDs) has been debated [15,16,17].

Depending on the underlying diagnostic method, SB is classified as possible, probable, or definite SB [1]: possible SB is based on a positive self-report only, probable SB is based on a positive clinical inspection, with or without a positive self-report, and definite SB is based on a positive instrumental assessment with or without a positive self-report and/or a positive clinical inspection. Non-instrumental methods register possible and probable SB and include self-assessment protocols and clinical examinations. In research, questionnaires are most commonly used to assess SB [18]. Clinical examination includes medical history via interview and a systematic examination of the teeth, soft and hard tissue, as well as the masticatory muscles [19]. Instrumental methods encompass extraoral and intraoral devices that measure current SB activity. Extraoral devices involve portable electromyography (EMG) measurements, which record the current activity of the masticatory muscles with high sensitivity and specificity (e.g., Bruxoff: sensitivity from 92.3% to 100% and specificity from 76% to 91.6% compared to gold-standard diagnosis) [20,21,22]. Advanced measurement includes PSG, which records sleep parameters in addition to masticatory muscle activity. This method is considered the gold standard for determining SB, even though it is time-consuming and expensive [23,24]. Although it is not mandatory to examine SB activity via audio and video recording, diagnostic accuracy rises with its use [25]. SB activity can be operationalized and summarized in episodes (SB-specific EMG signals per hour; SB index) [24]. Intraoral devices like thin sheets on the surface of teeth measure grinding patterns of SB activity. A recent study validated a novel diagnostic sheet with a fully automated analyzing software, calculating the amount of SB activity with a sensitivity of 100%, a specificity of 80%, and an area under the curve of 0.88 [26].

SB can thus be determined in different ways. Research results show that the measured relationship between SB and psychological distress varies depending on the type of SB classification. Possible SB correlates positively with both subjective and objectively measured stress (cortisol in saliva) [27,28]. Probable SB correlates positively with manic, depressive, and anxious symptoms, as well as with negative coping strategies or less positive coping strategies [29,30,31,32,33]. Objectively measured stress (salivary cortisol) also positively correlates with probable SB [29,32]. Looking at definite SB, there is no significant association between the number of SB episodes and anxiety (measured by the State-Trait Anxiety Inventory) and the expression of stress coping mechanisms [34]. Moreover, there is no significant difference between subjects with and without definite SB and chronic stress [35]. Depression, as measured by the Beck Depression Inventory, does not significantly correlate with the presence of definite SB and does not correlate with the number of SB episodes [10]. In contrast, Azevedo et al. show that subjects with definite SB have significantly higher anxiety levels [36]. The present study investigated general psychological distress, which was assessed by means of a self-assessment questionnaire.

In summary, it is still unclear to what extent SB correlates with psychological distress. One possible explanation for these differences could lie in the choice of diagnostic procedure. Therefore, the aim of the present study was to analyze SB’s relationship with psychological distress more in depth by applying the three recognized and, additionally, a novel diagnostic procedure for the estimation of SB in the same sample. The following questions were investigated in this study: (1) Are the procedures consistent with each other in the distribution of SB and non-SB diagnoses?; (2) Is there an association between the parameters of instrumental methods (PSG, diagnostic sheet) and psychological distress?; (3) Is there an association between the parameters of non-instrumental methods (self-report, clinical examination) and psychological distress? The null hypotheses are that there is no agreement between the diagnostic procedures and that there are no correlative relationships between psychological distress and SB diagnosed using any of these procedures.

## 2. Materials and Methods

The data analyzed in the present case–control study were secondary outcome variables from a previous monocentric validation study [26]. This study was conducted from May 2019 to July 2020. A total of four diagnostic procedures or classifications for SB were compared: (1) self-report, (2) clinical examination according to the criteria of the International Classification of Sleep Disorders (third edition, revised text; ICSD-3 TR), (3) PSG, and (4) diagnostic sheet combined with an analyzing software. The diagnostic procedures are described in more detail below. Psychological distress was assessed by a self-assessment instrument. Sample size calculation was based on the original study project [26]. The goal of the project was to validate a novel method, so the sample size planning was predetermined. The result of the power analysis based on the specific research question was *N* = 42. Considering a drop-out rate of 20%, a sample size of *N* = 50 subjects was targeted.

### 2.1. Subjects

Potential subjects were recruited via announcements at University Hospital Düsseldorf, Heinrich-Heine University, and the University of Applied Sciences Düsseldorf as well as via the institute’s own website and social networks. However, according to the principles of good clinical practice, included students must not be in any way dependent on the principal investigators. Therefore, dental students had to be excluded from participating in this investigation. Participation was open to healthy adults between the ages of 20 and 50 years. General exclusion criteria that would preclude a healthy general condition were as follows: severe mental disorder, abuse of or dependence on drugs or medications, central nervous system and/or peripheral nervous system disorders, and other serious physical or systemic illnesses, such as cardiovascular disease, autoimmune disease, respiratory insufficiency, or active inflammation or malignancy, as assessed by a medical history questionnaire. Pregnant and breast-feeding women were also excluded from participation. Other dental exclusion criteria were as follows: more than two missing molars (excluding third molars), the presence of a removable prosthesis or extensive prosthetic restorations, fixed orthodontic treatment, the presence of gross malocclusion (i.e., open anterior bite), and the presence of TMDs requiring treatment. In addition, no dental functional therapy may have taken place in the last six months. This includes, for example, the regular use of an occlusal splint or the application of physical therapy for the management of TMDs.

Dental and other inclusion and exclusion criteria were checked by one trained dentist within a clinical interview and thorough dental examination as published elsewhere [26]. The latter also included a verification of signs and symptoms of TMDs according to the German version of Research Diagnostic Criteria for TMDs [37]. The presence of TMD symptoms (not requiring treatment) was managed as a covariate, but was not an exclusion criterion, because this study focused on SB. At this dental appointment, impressions of the maxilla and mandible were taken for documentation purposes and for the fabrication of a diagnostic sheet. The presence of high psychological distress, which increases the likelihood of the occurrence of a mental disorder, was assessed with the SCL-90-S [38]. If subjects had a significantly high score (GSI: T score ≥ 63 or 2 scales: T score ≥ 63, respectively), they were excluded from participation in the study.

### 2.2. Sleep Bruxism Assessment

The examinations were carried out separately at four measurement points that were independent from each other. The subjects were to undergo the various measurements within a maximum of 28 days. First, to determine possible SB, subjects answered the question “Do you grind your teeth during sleep?” in a general self-evaluation clinical history protocol (screening questionnaire). Second, for the assessment of probable SB, the criteria of the ICSD-3 TR were determined via clinical interview and physical examination [39]: (A) the presence of repetitive jaw muscle activity characterized by grinding or clenching of the teeth in sleep, plus (B) the presence of one or more of the following clinical symptoms or signs consistent with the above reports of tooth grinding or clenching during sleep: (1) abnormal tooth wear; (2) transient morning jaw muscle pain or fatigue or temporal headache. The presence of occlusal tooth wear to at least the extent of dentin exposure was measured [40]. The following basic parameters were recorded during PSG: electrical brain activity (electroencephalography) measured with six different channels (F4-M1, C4-M1, O2-M1 with reserve electrodes on the opposite hemisphere), eye movements (electrooculography), heart rate (electrocardiography), respiratory effort with piezoelectric sensors, respiratory flow with nasal dynamic pressure sensors, leg movement (EMG), muscle tone (EMG), arterial oxygen saturation with pulse oximetry, and body position with movement sensors [41]. Moreover, the presence of masseter hypertrophy was examined upon voluntary clenching [42].

Third, for definite SB, data were collected via PSG using a SOMNOscreen (SOMNOmedics, Randersacker, Germany) with audio–video recording. Measurements were performed on two consecutive nights in the subjects’ home environment [39]. Two trained research associates prepared the measurement by a standardized protocol [43]. Moreover, EMGs of the masseter muscle and anterior temporalis muscle were recorded bilaterally. One trained scientist analyzed sleep data with the software DOMINO (version 2.9.0, SOMNOmedics, Randersacker, Germany). Sleep stages were scored manually using the guidelines of the American Academy of Sleep Medicine [44]. Definite SB was determined when the following research diagnostic criteria were met [24]: number of SB episodes/sleep hour (SB index) > 4, number of bursts/sleep hour > 25, number of SB episodes with teeth grinding sound > 1 (at least 1). A phasic episode is characterized by at least three muscle contractions (bursts) with a duration of 0.25 s to 2.00 s and visibly separated by episodes without activity. A sustained muscle contraction of at least 2.00 s duration characterizes a tonic episode. Between two episodes, there must be an episode of inactivity lasting at least 3.00 s to consider them separate episodes. If this episode of inactivity is missing between a phasic and tonic episode, it is called a mixed episode. A muscle contraction is scored when it exceeds 20% of the individual average amplitude of masticatory muscle activity under maximal voluntary clenching (MVC) [24]. MVC was measured on site before the main measurement. Of the two consecutive nights, the one with the highest SB index was chosen for further sleep data analysis. Sleep-related oromotor activity or EMG activity related to respiratory and other events were excluded from analysis via audio and video verification in order to include only SB-related events [25].

At the fourth and last visit, the diagnostic sheet with specific analyzing software was used. It was a 0.5 mm thin sheet made of biocompatible Terlux 2802 HD and consisted of five layers of different colors [26]. Each subject had to wear it on the maxilla for five consecutive nights. Nocturnal teeth grinding led to colored abrasion marks on the surface, which were analyzed fully automatically by a specific software. The outcome is the so-called pixelscore, which operationalizes the strength of SB activity. The higher the pixelscore, the stronger the nocturnal grinding activity.

### 2.3. Assessment of Psychological Distress

Psychological distress was assessed using the German version of the SCL-90-S self-report instrument [38]. Psychological distress was assessed using nine scales: somatization, obsessive–compulsive, interpersonal sensitivity, depression, anxiety, hostility, phobic anxiety, paranoid ideation, and psychoticism. The subjects completed the questionnaire using the paper-and-pencil method and the questionnaire analysis was computer-assisted. Standardized T-scales published in the manual were considered for the exclusion of test subjects. For the main analysis, the raw scores of the questionnaires were summed up and calculated according to the manualized procedure. In the present study, only the GSI, as a sum value for the instrument, was analyzed. It can assume a value between 0 and 4, with higher values reflecting higher psychological distress.

### 2.4. Outcome Variables

The outcome variables of the present study are summarized in Table 1.

### 2.5. Data Analysis

Descriptive methods were used for sample description. These included mean values (Ms) and standard deviations (SDs). In addition, frequency data were reported in absolute numbers and relative frequency data in percentages. Tests for group differences (e.g., Student’s *t*-test) were calculated as a function of scale level and distribution to test for systematic influences due to covariates such as age, gender, and education. The degree of agreement between the procedures was tested using Fleiss’ and Cohen’s kappa. The calculated values were interpreted according to standards [45]. If the kappa statistic is <0.00, it is to be interpreted as poor; between 0.00 and 0.20, as slight; between 0.21 and 0.40, as fair; between 0.41 and 0.60, as moderate; between 0.61 and 0.80, as substantial; and between 0.81 and 1.00, as almost perfect. Group comparisons and correlation analyses were performed to examine the association between psychological distress and SB. The choice of method (Pearson, Spearman, biserial) was based on the scale level and the distribution of the variables under investigation. Correlation coefficients were interpreted according to standards [46]. A coefficient between 0.1 and 0.3 is interpreted as a weak, a coefficient between 0.3 and 0.5 as a medium, and a coefficient >0.5 as a strong correlation. Before performing the statistical analysis, the prerequisites for each statistical test were checked. Shapiro–Wilk tests and visual inspections of quantile–quantile plots were performed to test for normal distribution. All calculations were performed using the R programming language and R Studio software (v. 4.2.1, RStudio Team, Boston, MA, USA). To avoid alpha error accumulation when performing multiple testing, the false discovery rate (FDR) was controlled [47]. The significance level was set at *p* = 0.05 for all calculations.

## 3. Results

A total of 45 participants were recruited for this study, with *n* = 22 (48.89%) female and *n* = 23 (51.11%) male participants. Ages ranged from 21 to 46 years with M = 26.40 (SD = 4.44). The proportion of students was 78%. All participants were classified as SB or non-SB according to four different diagnostic procedures. The distribution of the frequency of SB and non-SB depending on the diagnostic procedure is shown in Table 2.

Table 3 shows the descriptive statistics of specific SB parameters according to the respective diagnostic procedures. The two variables, SB after self-report (screening) and SB after self-report (interview clinical examination), matched in *n* = 39 subjects (87.00%).

### 3.1. Agreement between Diagnostic Procedures

In the first step, it was tested whether the procedures agree with each other in the distribution of SB and non-SB diagnoses. The agreement among all procedures was Fleiss’ κ = 0.39 with z = 6.33, *p* < 0.01, which can be interpreted as moderate [45]. In particular, the distribution of SB (*n* = 10; 22.22%) and non-SB (*n* = 35; 77.78%) assessed using PSG was very unequal, whereas self-reported SB (*n* =21; 46.67%) and non-SB (*n* = 24; 53.33%) were almost equally distributed. In *n* = 23 (51.11%) of the subjects, the diagnosis (SB or non-SB) was the same after all four different procedures. For the remaining subjects, allocation into the SB or non-SB group was heterogeneous depending on the applied diagnostic procedure (e.g., one subject was assigned as SB using self-report, but the same subject was assigned as non-SB using the three other procedures). Table 4 shows the agreement of the methods compared with the gold standard, PSG. Here, the degree of agreement was calculated using Cohen’s kappa. The agreement between PSG and the clinical examination as well as the self-report is to be interpreted as moderate [45]. The agreement between PSG and the diagnostic sheet was substantial [45].

### 3.2. Group Comparison between SB Diagnosis and Psychological Distress

The values of SCL-90-S GSI showed a right-skewed distribution for the total sample, with M = 0.20 and SD = 0.26 (median = 0.11, range 0.02–1.19). Data were not normally distributed (W = 0.64, *p* < 0.001) after both visual inspection of the quantile–quantile plot and calculation of the Shapiro–Wilk test. Figure 1 shows the distribution of the GSI data by diagnosis using the four different procedures in the form of boxplots.

Based on the data in the boxplots, it can be seen that the location parameters deviate. Due to the outliers, the mean values of the GSI are clearly shifted, whereas the median is almost identical in all groups regardless of the diagnostic method. As the data in the GSI are non-normally distributed, group comparisons were calculated using Mann–Whitney-U tests. The *p*-value was adjusted for multiple comparisons using the FDR. Table 5 presents the descriptive statistics of the GSI depending on the diagnosis of SB (yes vs. no) according to the respective diagnostic procedure with statistical group comparison. None of the comparisons showed a statistically significant result.

### 3.3. Correlational Analyses of Specific SB Parameters and Psychological Distress

In the next step, correlation analyses were performed to calculate, on the one hand, the correlation between the diagnosis of SB (dichotomous variable) depending on the procedure and the GSI. On the other hand, the correlation between specific diagnostic parameters (interval-scaled and dichotomous variables) and the GSI was examined. Table 6 shows the coefficients of the correlations between the diagnostic parameters and the GSI.

Spearman correlation coefficients (r_s_) or biserial correlation coefficients (r_b_) were calculated depending on the scale level. Spearman correlation coefficients were calculated and tested. Because the GSI was non-normally distributed, biserial correlation coefficients could not be statistically tested using the z-test [48]. Thus, no *p*-values are given. None of the Spearman correlation coefficients were statistically significant. The strength of the correlation coefficients was also largely low, except for the correlation between self-reported SB in the clinical interview and the GSI (r_b_ = 0.44). This means that self-reported SB in the clinical interview is associated with high psychological distress. This was not true for the other parameters, as other dental parameters correlated only slightly with the GSI (e.g., presence of masseter hypertrophy, r_b_ = 0.12). The frequency of masticatory muscle contractions (SB index, r_s_ = 0.05) was almost not at all correlated and pixelscore was even weakly negatively correlated with the GSI (r_b_ = −0.11).

## 4. Discussion

Based on statistical group comparison, psychological distress, as measured by the GSI of the SCL-90-S, did not differ significantly between subjects with or without possible, probable, or definite SB. A more in-depth analysis showed that there is a medium positive but non-significant correlation between self-reported SB assessed in the clinical interview and psychological distress. There is a near-zero correlation between PSG parameters, such as the SB index, and psychological distress. That is, neither masticatory muscle activity nor audible grinding sounds are related to psychological distress. A weak negative but non-significant correlation exists between the current level of SB activity, measured using a new diagnostic procedure with fully automated evaluation software, and psychological distress. The diagnostic procedures are moderately in agreement with each other. Individual comparisons show that agreement between self-reported as well as clinically determined SB and definite SB is low. However, agreement between the newly developed intraoral device (the diagnostic sheet) and PSG was substantial. The low agreement especially between clinically found SB and instrumental methods is in line with other studies [49,50].

The results concerning psychological distress and SB are not in line with other studies, which found a significant difference in SCL-90-R scores and subscales (anxiety) between subjects with and without possible and probable SB [51,52]. It should be noted here that the sample of Ahlberg et al. was much larger (*N* = 750) and only irregular shift workers were recruited [52]. Bayar et al. included only patients from a military hospital [51]. The sample of the present study mainly comprised student subjects, which may make it difficult to compare it with the other studies. To the authors’ knowledge, no further study has yet examined the difference between probable SB and non-SB via the SCL-90-R. However, researchers have been able to show that individuals with probable bruxism show more stress and poorer mental condition, both subjectively and objectively [13,29,30,31,32,33]. Based on these findings, it seems contradictory that there is no difference between probable and non-SB inventory in the present study.

Few studies address the relationship between current SB activity measured using instrumental methods and psychological distress measured using the SCL-90-S. Regarding instrumentally measured SB activity, the present result contradicts that of Shen et al., who found both a significant difference and significant positive associations between the number of SB episodes and scores on the SCL-90-R [53]. However, the findings of Manfredini et al. corroborate the pattern of the results of the present study [54]: based on a meta-analysis, it was shown that non-instrumentally measured SB parameters are more likely to be related to psychosocial factors than instrumentally measured SB parameters. In the present study, on a descriptive level, correlations are also higher between non-instrumental SB parameters (e.g., presence of self-reported SB) and psychological distress, compared with instrumentally measured SB parameters (e.g., SB index) and psychological distress.

Some general and methodological limitations should be mentioned. The SCL-90-S was only collected at screening. The time interval to PSG recording was up to two to three weeks for the majority of subjects. However, the SCL-90-S has largely good test–retest reliability, indicating that originally recorded values are stable over time [38]. It can be assumed that the variance in psychometric measures is limited in the sample. This can be attributed to the fact that the SCL-90-S initially served as an instrument to exclude potential subjects who were suspected of having a mental illness. The uneven distribution of SB subjects (e.g., definite SB, *n* = 10; non-SB, *n* = 35) and the generally small sample size could complicate the interpretation of diverse results.

A major advantage of the present study is the application of four different diagnostic approaches (two non-instrumental and two instrumental ones) for SB assessment in the same sample at different time points. The results provide the idea that the choice of diagnostic method possibly influences the association between SB and psychological distress. It could be speculated that SB can be considered a construct, with the different diagnostic procedures capturing variant aspects of this construct. One possible implication is that the underlying muscle and grinding activity has no or even a slight negative relationship with psychological distress. Another consideration might include that self-reported SB, whether via a questionnaire or at the dentist’s appointment, also reflects general distress.

Considering these limitations, suggestions for further research projects emerge in order to take a more in-depth look at the relationship between SB and psychological distress. First, measuring psychological distress and SB activity close in and over time would be useful to better capture the relationship between changes in psychological distress and changes in SB activity. Second, in order to increase sample size, investigations applying simple devices like EMG or an intraoral apparatus (e.g., diagnostic sheets) would be more practical [20,26,55,56]. Especially for the diagnostic sheet, the data of the present study provide a very good agreement concerning the determination of SB compared with the gold standard [25]. It might be interesting to see if other factors cause individuals to identify SB in themselves. Detecting SB by oneself is difficult because the events happen during sleep. Concomitant symptoms, such as stiffness in the masticatory muscles or fatigue, could also be caused by other phenomena, such as awake bruxism or TMDs. Accordingly, for future research purposes, it would be interesting to assess the influence of AB, SB, and TMDs on mental stress, preferably using instrumental methods.

## 5. Conclusions

The agreement of SB and non-SB diagnoses between specific instrumental and non-instrumental procedures is low. It can be assumed that the procedures operationalize different aspects of SB. The association between psychological distress and SB measured with instrumental procedures (PSG, diagnostic sheet) is descriptively low. The association between psychological distress and SB measured with non-instrumental methods (self-report, clinical examination) is, descriptively, slightly stronger and positive. In summary, considering the limitations, the following can be suggested for clinical practice and research: when SB is measured non-instrumentally, the association with psychological distress is stronger than after instrumental measurement. Nonetheless, if psychological distress is reported or recognized in the context of possible or probable SB, practitioners should take this seriously and psychotherapeutic support can be recommended.

## Figures and Tables

**Figure 1 jcm-13-00638-f001:**
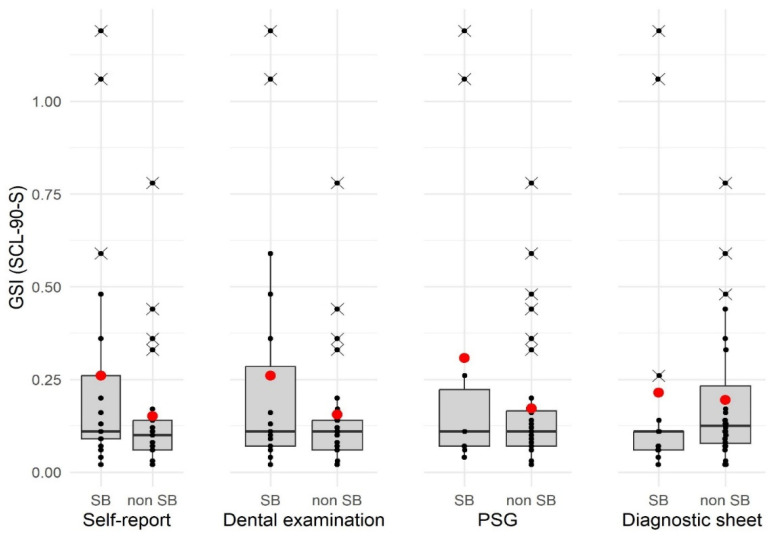
Boxplots of GSI (SCL-90-S) data divided by bruxism classification and diagnosis procedure. The red dots indicate the mean values. The black dots indicate individual data values. Data values marked with a cross are outliers.

**Table 1 jcm-13-00638-t001:** Summary of outcome variables in the present study, divided by diagnostic procedure.

Diagnostic Procedure	Method	Outcome	Classification
Self-report	Single question “Do you grind your teeth during sleep?” in the screening	SB self-report (screening) (y/n)	Possible SB
Clinical examination	Based on ICSD-3 TR: Inspection of teeth Anamnestic interview	SB self-report (interview) (y/n) SB external report (y/n) Tiredness/stiffness (y/n) Masseter hypertrophy (y/n) Number of teeth with abnormal attrition	Probable SB
	Examination of masseter muscle		
PSG	Analysis of physiological sleep data including muscle activity of masseter muscle	SB index Number of episodes with grinding sounds index	Definite SB
Diagnostic sheet and software	Fully automated, computer-based analysis of wear on the diagnostic sheet	Pixelscore	-
Psychometric self-evaluation	SCL-90-S	GSI	-

Note: y = yes; n = no.

**Table 2 jcm-13-00638-t002:** Distribution of SB and non-SB across diagnostic procedures (*N* = 45).

Diagnostic Procedure	SB *n* (%)	Non-SB *n* (%)
Self-report	21 (46.67%)	24 (53.33%)
Clinical examination	20 (44.44%)	25 (55.56%)
PSG	10 (22.22%)	35 (77.78%)
Diagnostic sheet and software	17 (37.78%)	28 (62.22%)

**Table 3 jcm-13-00638-t003:** Descriptive statistics of specific SB parameters as a function of diagnostic procedure.

	**SB Diagnosis Based upon PSG**			
	**SB (*n* = 10)**		**Non-SB (*n* = 35)**	
**Specific Parameter**	**M**	**SD**	**M**	**SD**
SB index	8.40	4.56	2.40	0.96
Episodes with grinding sounds index	3.61	4.77	0.26	0.37
	**SB Diagnosis Based upon Clinical Examination**			
	**SB (*n* = 20)** ***n* (%)**		**Non-SB (*n* = 25)** ***n* (%)**	
SB self-report (interview):				
Yes (*n* = 15)	15 (75.00%)		0 (0.00%)	
No (*n* = 30)	5 (25.00%)		25 (100.00%)	
SB third-party report:				
Yes (*n* = 14)	14 (70.00%)		0 (0.00%)	
No (*n* = 31)	6 (30.00%)		25 (100.00%)	
Feeling of tiredness/stiffness:				
Yes (*n* = 15)	13 (65.00%)		2 (8.00%)	
No (*n* = 30)	7 (35.00%)		23 (92.00%)	
Masseter hypertrophy:				
Yes (*n* = 14)	10 (50.00%)		4 (16.00%)	
No (*n* = 31)	10 (50.0%)		21 (84.00%)	
	**M**	**SD**	**M**	**SD**
Number of teeth with abnormal attrition	7.90	6.68	6.00	5.40
	**SB Diagnosis Based upon Diagnostic Sheet**			
	**SB (*n* = 17)**		**Non-SB (*n* = 18)**	
	**M**	**SD**	**M**	**SD**
Pixelscore	1274.23	735.72	168.69	139.00

Note: The bold headings divide the descriptive statistics of the different diagnostic classifications and the distribution of values of the specific parameters.

**Table 4 jcm-13-00638-t004:** Agreement of diagnoses of all procedures with diagnosis using the gold-standard reference, PSG (*N* = 45).

	Diagnosis Based upon PSG		
Diagnoses Based upon Other Procedures	SB (*n* = 10) *n* (%)	Non-SB (*n* = 35) *n* (%)	Statistics
Self-report:			
SB (*n* = 21)	7 (70.00%)	14 (40.00%)	κ = 0.22
non-SB (*n* = 24)	3 (30.00%)	21 (60.00%)	
Clinical examination:			
SB (*n* = 20)	7 (70.00%)	13 (37.14%)	κ = 0.24
non-SB (*n* = 25)	3 (30.00%)	22 (62.86%)	
Diagnostic sheet and software:			
SB (*n* = 17)	10 (100.00%)	7 (20.00%)	κ = 0.64
non-SB (*n* = 28)	0 (0.00%)	28 (80.00%)	

**Table 5 jcm-13-00638-t005:** Descriptive statistics and comparison on group differences of GSI as a function of SB diagnosis by respective procedure.

	GSI (SCL-90-S)				
Diagnosis SB	M	SD	Median	Statistics	*p* ^†^
Self-report:				W = 190.50	
Yes (*n* = 21)	0.26	0.32	0.11		0.400
No (*n* = 24)	0.15	0.17	0.10		
Clinical examination:				W = 215.00	0.572
yes (*n* = 20)	0.26	0.33	0.11		
No (*n* = 25)	0.16	0.17	0.11		
PSG:				W = 172.00	
Yes (*n* = 10)	0.31	0.44	0.11		0.945
No (*n* = 35)	0.17	0.17	0.11		
Diagnostic sheet and software:				W = 293.00	0.400
yes (*n* = 17)	0.21	0.35	0.11		
No (*n* = 28)	0.20	0.19	0.12		

Note: ^†^ significance value controlled with FDR.

**Table 6 jcm-13-00638-t006:** Correlation analyses of specific SB parameters and SCL-90-S GSI.

	GSI (SCL-90-S)		
Parameter	r_b_	r_s_	*p* ^†^
Self-report:			
SB self-report (screening)/Diagnosis of SB	0.27	-	-
Clinical examination:			
Diagnosis of SB	0.26	-	-
SB self-report (interview)	0.44	-	-
SB third-party report	0.20	-	-
Feeling of tiredness/stiffness	0.24	-	-
Masseter hypertrophy	0.12	-	-
Number of teeth with attrition	-	0.22	0.580
PSG:			
Diagnosis of SB	0.31	-	-
SB index	-	0.05	0.715
Episodes with grinding sounds index	-	−0.06	0.715
Diagnostic sheet:			
Diagnosis of SB	0.05	-	-
Pixelscore	-	−0.11	0.715

Note: r_b_ = biserial correlation coefficient, r_s_ = Spearman correlation coefficient, ^†^ significance value controlled with FDR.

## Data Availability

The datasets generated during and/or analyzed during the current study are available from the corresponding author on reasonable request.

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
