# Peer review of "Association between Psychological Distress and Possible, Probable, and Definite Sleep Bruxism—A Comparison of Approved Diagnostic Procedures"

_jcm, 2024, doi:10.3390/jcm13020638_

Round 1

Reviewer 1 Report

Comments and Suggestions for Authors

Thank you for the opportunity to review this manuscript. The topic is highly interesting but the study design and sample size do not allow for concluding. I found it too speculative. On the other hand, I found a lot of bias. It is important to explain the biases, and restructure certain parts of the discussion; add all the existing limitations in a specific section at the end of the discussion. Conclusions should respond specifically to the objectives and not be reiterative.

Line 30. Please add the last definition of bruxism and specifically the SB's last definition: “Sleep bruxism is a masticatory muscle activity during sleep that is characterized as rhythmic (phasic) or non-rhythmic (tonic) and is not a movement disorder or a sleep disorder in otherwise healthy individuals…”

Line 47. Please note that the physical exploration includes more than the tooth wear as an isolated sign; for example the soft and hard tissue exploration, masticatory muscles…

Line 90. The physical exploration included the DC/TMD protocol ?. Please, add all the sample selection and examination protocols.

Line 91. Please add the reference and specify the criteria used for the PSG diagnosis.

Line 101. Why did you decide the 21-50 range of age as inclusion criteria?

Line 108. Why two or more loss molar was an exclusion criterion? You didn’t specify the tooth wear test was used and did not specify if excluded the mixed wear and erosion wear…Please describe it. 

Line 111. Please make sure that you have specified for the first time earlier in the text “Temporomandibular disorders (TMD)”.

Line 118. I don’t understand if the TMD was managed as a covariate or was excluded. It is confusing, please explain this better.

Line 129. What is the “specific questionnaire”? Please, add the reference and details.

Line 138. Please specify whether home PSG is monitored or not.

Line 141. Please specify if the PSG records were automatic or manually analyzed.

Line 151. Please clarify if used the 20% of increase of baseline activity or the 20% of the maximum voluntary contraction. If was the last, you need to specify that…For example: “a previous MVC was registred”. Did you the MVC previous register? Could explain that?

The PSG sleep bruxism diagnosis as the gold standard should be better explained.

Did you exclude the Sleep-related oromotor activity (OMA)?...

Did you find other sleep disorders such as obstructive sleep apnea during the analyses?... How was quantified the electromyography (EMG) activity after the respiratory events?...

I understand that only the RMMA was accounted for, so all the MMA (not rhythmic activities) was not accounted?

The RMMA that does not correspond with the classical autonomic cascade of RMMA taken into account? I mean phasic episodes without a previous heart rate increase…

Line 157. I understand that only the grinding noise was taken into account for the PSG analysis, so can be predictable agreement with the pixelscore.

However, the EMG tonic activity corresponding with clenching or without sound was not counted along the PSG analysis…The last consensus indicated that the presence of noise is recommendable but not mandatory… And it is an important bias to the final analysis data. Remarking this in the PSG analysis methods section and then at the discussion section is fundamental. It is important to note that the pixelscore maybe size objective forces or teeth contact, but it is not EMG activity.

If include that in the discussion, consider adding the last reference of the ICSD-3 TR, which includes the clenching (“A. The presence of repetitive jaw-muscle activity characterized by grinding or clenching of the teeth in sleep. B. The presence of one or more of the following clinical symptoms or signs consistent with the above reports of tooth grinding or clenching during sleep: 1. Abnormal tooth wear. 2. Transient morning jaw muscle pain or fatigue, or temporal headache”).

https://aasm.org/wp-content/uploads/2023/05/ICSD-3-Text-Revision-Supplemental-Material.pdf

Line 216. Please, revise the results: “The diagnoses matched for n = 23 (51.11 %) subjects according to all four procedures. Table 3”.

However, in table 3 there are 24 subjects (7+7+10) with positive SB diagnosis.

Line 227. Please define all the variables in the methods section. What is the difference between self-report (screening) and self-report (interview clinical examination)?

Following the last international consensus (2018) recommendations…

Possible SB – Self-report

Probable SB – Physical exploration (with or without self-report), and not only the tooth wear examination…

Definitive SB – EMG or PSG with or without noises, PSG is the gold standard.

Line 249-250. This limitation and others should be included in a specific “limitations section” at the end of the discussion.

Line 276. There is not sufficient data for this affirmation, in any case, you should specify “…that increased rhythmic masticatory muscle activity during sleep…” or “… that increased SB activity …”, because you cannot talk about general muscle activity if you did not account the MMA.

Line 300. Please check this article 10.3390/ijerph20032452. I think that can be useful to compare with your results in the discussion.

Line 303. Awake bruxism (AB) was not evaluated, so maybe some subjects did not have SB but had AB, or both SB and AB. This point deserves to be commented on in the discussion.

Line 328. “…for SB assessment in the same sample at different time points”. Please, explain the chronology and the proofs more in detail in the methods section. For example: the PSG recordings and the new diagnosis tool were simultaneously? The self-report was the same week, day, month…

Line 352. The conclusions must respond specifically to the objectives, not be reiterative. Please rephrase this part to make it clearer and more concise.

Author Response

Dear Reviewer,

thank you very much for your comments. Please see the attachment, which contains the point-by-point responds.

Kind regards,

Nicole Walentek

Reviewer 2 Report

Comments and Suggestions for Authors

This study has some scientific interest to understand the performance of the tools used. At present, however, it is unclear how useful the findings are to clinical or public health practice. My specific comments are given below.

Introduction:

Lines 62-75: This section contains information on various psychological aspects, but it is unclear how these are related to the focus (psychological distress) of the manuscript. A clearer description of what is meant by psychological distress is warranted.

Methods:

The authors mention of the quasi-double-blind design could be misleading as, apparently, it is not the design used for this analysis.

The rationale for some exclusion criteria such as systemic illnesses, such as cardiovascular disease, autoimmune disease, respiratory insufficiency etc is not clear.

Whether TMD was excluded or used as a covariate is also not clear – the authors have indicated both these at two different places.

A brief description of how SCL 90-S was scored is needed.

Results:

The description related to exclusion of students is unclear, and should come under the methods. The basic characteristics of the participants have not been presented.

The description of results under section 3.1 is not clear enough. It would be better to indicate the performance of other tests against the gold standard and then the comparison between the indicator tests.

The data relevant to Table 4 are more descriptive and could come before the comparison between the tests. It is also unclear how some of the content in Table 4 is relevant to the research questions.

Table 6 is unclear. What are rb and rs? What does the p+ mean (instead of )?

Discussion:

The authors have compared their results with those of others, but have not attempted to discuss any clinical/practice implications of these findings.

Comments on the Quality of English Language

The writing can be improved to provide better clarity and conciseness. 

Author Response

(The authors gave the same response as above.)

Reviewer 3 Report

Comments and Suggestions for Authors

This study compared the association between psychological distress and self-reported, clinically assessed, and polysomnography-diagnosed sleep bruxism (SB). The results were quite interesting for the study of sleep bruxism. I have some considerations to make regarding the study:

Abstract: Specify the study type.

Introduction: Include the null hypothesis.

Material and Methods: Why were participants aged between 20 and 50 years chosen for the study?

Material and Methods: Why were two polysomnography (PSG) sessions used for SB diagnosis?

Material and Methods: Provide more details on the sample calculation (effect size, test power, software used).

Material and Methods: Was posterior crossbite not considered an exclusion criterion?

Material and Methods: How was the presence of masseter hypertrophy evaluated for SB diagnosis?

Author Response

(The authors gave the same response as above.)

Round 2

Reviewer 1 Report

Comments and Suggestions for Authors

Thank you for improving the manuscript. I think it is an interesting work and is in line with all the evidence from scientific research in the field.

I just wanted to comment that the reference to the updated ICSD-3 (ICSD-3 TR) is missing and should be included. Perhaps it is advisable to add in line 159 more technical specifications of the PSG used (number of EEG leads, EOG... If it is a PSG type II, specify it).

Congratulations on your paper and your effort.

Author Response

Dear Reviewer,

Thank you very much for your kind reply. We have gladly taken your comments into consideration. Enclosed you will find our answers. 

Thank you very much and kind regards
Nicole Walentek
